# Identification of Genetic Causes in Mayer-Rokitansky-Küster-Hauser (MRKH) Syndrome: A Systematic Review of the Literature

**DOI:** 10.3390/children9070961

**Published:** 2022-06-27

**Authors:** Varvara Ermioni Triantafyllidi, Despoina Mavrogianni, Andreas Kalampalikis, Michael Litos, Stella Roidi, Lina Michala

**Affiliations:** 11st Department of Obstetrics and Gynecology, ‘Alexandra’ General Hospital, National and Kapodistrian University of Athens, 80 Vasilissis Sofias Ave, 11528 Athens, Greece; andreas.kal@hotmail.com (A.K.); s_roidi@hotmail.com (S.R.); linamichala@med.uoa.gr (L.M.); 2Molecular Biology Unit, Division of Human Reproduction, 1st Department of Obstetrics and Gynecology, ‘Alexandra’ General Hospital, National and Kapodistrian University of Athens, 80 Vasilissis Sofias Ave, 11528 Athens, Greece; depy.mavrogianni@yahoo.com; 3Department of Obstetrics & Gynecology, Konstantopouleio General Hospital of Nea Ionia, 14233 Athens, Greece; mlitos@hotmail.com

**Keywords:** Mayer-Rokitansky-Küster-Hauser (MRKH) syndrome, Rokitansky, uterine aplasia, uterine anomalies, genetics

## Abstract

Mayer-Rokitansky-Küster-Hauser (MRKH) syndrome is a congenital condition characterizing females with absence of the uterus and part of the vagina. Several genetic defects have been correlated with the presence of MRKH; however, the exact etiology is still unknown due to the complexity of the genetic pathways implicated during the embryogenetic development of the Müllerian ducts. A systematic review (SR) of the literature was conducted to investigate the genetic causes associated with MRKH syndrome and Congenital Uterine Anomalies (CUAs). This study aimed to identify the most affected chromosomal areas and genes along with their associated clinical features in order to aid clinicians in distinguishing and identifying the possible genetic cause in each patient offering better genetic counseling. We identified 76 studies describing multiple genetic defects potentially contributing to the pathogenetic mechanism of MRKH syndrome. The most reported chromosomal regions and the possible genes implicated were: 1q21.1 (*RBM8A* gene), 1p31-1p35 (*WNT4* gene), 7p15.3 (*HOXA* gene), 16p11 (*TBX6* gene), 17q12 (*LHX1* and *HNF1B* genes), 22q11.21, and Xp22. Although the etiology of MRKH syndrome is complex, associated clinical features can aid in the identification of a specific genetic defect.

## 1. Introduction

The Mayer-Rokitansky-Küster-Hauser (MRKH) syndrome, or Müllerian aplasia, is a syndrome that affects females and is characterized by the absence of the uterus and the upper part of the vagina. These individuals have a normal karyotype (46, XX) and usually normal ovarian function [1]. It is divided into two types: Type I is characterized by uterovaginal aplasia, while Type II is additionally related to extragenital anomalies, most commonly renal (30–40%), skeletal, ear, and cardiac anomalies [2,3]. The reported incidence rate of MRKH syndrome is around 1:5000 live female births and, due to this rarity, it is poorly investigated [2,4]. In most cases, MRKH syndrome is diagnosed due to the presence of primary amenorrhea. The impact of the MRKH diagnosis and the associated psychological burden on young girls is significant [5]. The treatment of the syndrome includes vaginal dilation or, in case of failure or non-compliance with treatment, operative creation of a neovagina. Concerning the fertility of MRKH individuals, surrogacy is an option; however, uterine transplantation has been recently introduced [6].

Embryologically, the female reproductive system in humans derives from the Müllerian or Paramesonephric Ducts (MDs), which give rise to the uterus, cervix, and the upper two-thirds of the vagina at around the fifth to sixth week of gestation [1,7,8]. Several gene defects can affect the embryogenetic pathways of the development of the female reproductive system and cause MRKH syndrome.

Evidence of the inheritance pattern of MRKH syndrome remains scarce, due to the fertility restrictions of MRKH patients in the past; hence, no family trees were available to study [9,10]. The majority of the cases are sporadic, though there have been reports of familial cases, with a recent increase in the latter due to the introduction of surrogacy and uterine transplantation [10,11]. First-degree relatives of MRKH patients seem to have a 1–5% risk of congenital uterine anomalies as in most multifactorial disorders [12]. The majority of the studies of familial cases suggest an autosomal dominant inheritance pattern limited to the female sex, implying that the genetic defect is typically inherited by the father [13]. Wolffian duct hypoplasia, or agenesis, and other defects such as renal anomalies, hearing impairment, and skeletal deformities have been reported in the males of these families, similar to MRKH patients [10,11,14]. Another finding of note is that some articles refer to the presence of discordant monozygotic twins with MRKH syndrome, implying that environmental factors, i.e., epigenetic changes, may play a role in gene expression affecting Müllerian duct development [15,16,17,18,19,20,21,22,23].

The aim of this study was to systematically review the available literature and to summarize all genetic defects that have been described in MRKH patients. In addition, this study aimed to present commonly studied genes in correlation with their associated clinical features in order to provide guidance to clinicians and geneticists in their efforts to identify a specific genetic defect in each patient. This information can aid in genetic counseling and lead to more favorable outcomes through the early detection of specific genetic defects in pregnancy and the possibility of gene therapy in the future. Ultimately, these insights could be of use in guiding further genetic studies on MRKH syndrome.

## 2. Materials and Methods

For the conduction of this systematic review, a protocol based on the Preferred Reporting Items for Systematic Reviews and Meta-Analysis (PRISMA) guidelines, was used, following the PRISMA assessment checklist [24]. The search terms for our research included: Mayer–Rokitansky–Küster–Hauser syndrome; Rokitansky; uterine aplasia; uterine abnormalities; and genetics. We searched four different databases: Pubmed, ScienceDirect, Scopus, and Web of Science. The search was performed in December 2021 and was updated on 25 May 2022. All studies that reported on the genome of human female patients with MRKH and/or CUAs from 1994 to 25 May 2022 were included; studies not published in English, studies on animals, non-research studies, non-genetic studies on MRKH or CUAs, and studies not concerning MRKH or CUAs were excluded. The study selection was conducted by two independent reviewers (V.T. and A.K.), while a third reviewer (L.M.) assisted in decision-making when there was a conflict of opinion. The retrieved articles were compiled and de-duplicated. Additional eligible studies were retrieved by hand searching the citations from all articles. All studies meeting the inclusion criteria were included in the review. For every eligible article, information regarding the date of publication, the main findings, and the number of patients and controls were recorded. The study did not involve contact with humans, so the need for ethical approval was waived. This review was not registered. The selection and screening process are presented in the PRISMA flowchart shown in Figure 1.

## 3. Results

A total of 162 articles were identified from all databases using the search strategy, of which 32 were duplicates. In total, 30 eligible studies were identified from the hand search of the citations of the articles. According to both our inclusion and exclusion criteria, in total, 76 studies were considered eligible and included in this SR. Table A1 presents all 76 studies, sorted by year of publication, along with the main results and the number of individuals with MRKH syndrome or CUAs and controls who were studied.

The most reported chromosomal regions and the possible genes implicated are: 1q21.1 (*RBM8A* gene), 1p31-1p35 (*WNT4* gene), 7p15.3 (*HOXA* gene), 16p11 (*TBX6* gene), 17q12 (*LHX1* and *HNF1B* genes), 22q11.21, and Xp22.

Table 1 presents the chromosomal regions most commonly implicated in MRKH syndrome and CUAs and the suspicious genes involved, as indicated by animal and human studies. This table also presents the clinical features associated with defects in the respective genetic locations, the main results of non-human studies regarding these chromosomal regions, and whether they are linked with Type I or Type II MRKH.

## 4. Discussion

In this review, we have thoroughly analyzed the studies examining the genetic causes of MRKH syndrome. We endeavored to present our findings comprehensively and aimed to help clinicians associate clinical presentations with specific genetic defects. The need for genetic advice has become increasingly important in recent years due to the introduction of surrogacy and, most recently, uterine transplantation. The information included in this review regarding the genetic cause and pathogenesis of MRKH syndrome could significantly improve the counseling offered to individuals with MRKH and their families.

Our search confirmed that the genetic background of MRKH is poorly studied [25,28]. Mice models with targeted mutagenesis identified multiple genes that affect the development and differentiation of the female reproductive system during embryogenesis (Table 1). According to these studies, a number of candidate genes have been proposed as the causative factor for MRKH syndrome in humans and have been analyzed using array-comparative genomic hybridization (CGH) and whole-genome sequencing (WGS). Many MRKH patients have been reported to carry chromosomal anomalies that affect multiple chromosomal regions.

Despite the myriad of sporadic gene variants found through our search, we have identified a recurring pattern of affected chromosomal locations. Most reported chromosomal regions with their most implicated genes are: 1q21.1 (*RBM8A* gene), 1p31-1p35 (*WNT4* gene), 7p15.3(*HOXA* gene), 16p11 (*TBX6* gene), 17q12 (*LHX1* and *HNF1B* genes), 22q11.21, and Xp22 [10,25,26,28,33,43,44,45,46,48,54,55,58,59,60,61].

1q21.1

Affected regions in 1q21.1, a well-known location in TAR syndrome cases (thrombocytopenia/absent radius), have been identified in patients with Müllerian malformations [25,26,27,28,30]. More accurately, variants of the *RBM8A* gene—which is located in this chromosomal region—have been proposed as the possible cause of MRKH syndrome and gonadal dysgenesis, as this gene mainly affects oocyte differentiation and determination of the primordial germ cells [28,29].

1p31-1p35

*WNT4* is important in MD development during embryogenesis. It plays a double role in the female gonad: it controls female development and prevents testes formation [28,62,63,64,65,66]. For this reason, when MRKH syndrome is combined with signs of hyperandrogenism, heterozygous variants of the *WNT4* gene may be considered. Moreover, folliculogenesis in affected women can also be disrupted because of the gene’s role in the development of the gonad [28,65].

7p15.3

Τhe *HOX* clusters belong to a large family of homeobox-containing genes. The *HOXA* genes affect the development of the female reproductive system, as has been indicated by human and animal studies, and are, therefore, considered to be strong candidates for MRKH syndrome [10,67]. Despite their central role in the formation of MD, variants of these genes have been identified in only a few MRKH patients and are of unknown significance [10,68].

16p11.2

Deletions in 16p11.2 have been associated with autism and other neurological disorders (i.e., epilepsy, seizures, and learning disabilities), as well as congenital uterine anomalies [28,33]. The *TBX6* gene, which is located in this region, encodes a transcription factor that affects the embryogenetic development and, more specifically, the differentiation of the mesoderm. Therefore, it is suggested to be a putative candidate for MRKH syndrome [9,12,27,28,30,34,36,37,38].

17q12

It is known that 17q12 is the most affected chromosomal location in MRKH syndrome [25,26,27,33,43,45]. Associated anomalies of genetic defects in this location include renal cysts, mild facial malformations, severe cognitive disabilities, and seizures [26,46].

Specifically, *LHX1* (*LIM* homeobox protein 1) and *HNF1B* (hepatocyte nuclear factor *1B*; also known as *TCF2*) genes seem to be important candidates based on the prevalence of their variants in MRKH patients and their established roles in the development of the reproductive and urinary system [28].

*LHX1* has been associated with MRKH type II and unilateral renal agenesis [25,34,44]. The gene influences CNS formation [69,70] and has also been described in MRKH patients with mild mental and learning disabilities [26,28,44,48].

Variants and deletions of the *HNF1B* gene are characteristic in renal cysts and diabetes, which may be explained by the expression of the *HNF1B* gene in the kidney and pancreas; *HNF1B* is also expressed in the Wolffian and Müllerian ducts and plays a central role in their formation [53]. Some researchers have reported variants of the *HNF1B* gene in familial cases of CUAs, often associated with kidney malformations and Maturity Onset Diabetes of the Young (MODY) [49,50].

22q11.21

Deletions in 22q11.21 are responsible for DiGeorge or Velocardiofacial Syndrome (DG/VCFS). This syndrome can be manifested with variable phenotypes, occasionally including CUAs; therefore, MRKH syndrome may be a part of DG/VCFS. Changes in the *TBX1* gene, which is located in 22q11.21, are considered to be responsible for DGS/VCFS. However, this gene has not been associated with MRKH. This finding suggests that other genes in this region may be responsible for the appearance of CUAs [25,26,27,28,33,56].

Another issue of interest is the lack of large cohort studies associating a single gene variant solely with MRKH type I or MRKH type II. This may be due to the fact that genes that affect MD development during embryogenesis can also affect the development of the urinary system, owing to their common origin. Moreover, the sample of individuals in most genetic studies consists of both individuals with MRKH type I and individuals with MRKH type II; consequently, *LHX1* and *GREB1L* genes can affect the development of both the reproductive and urinary systems and, therefore, have been correlated mainly with MRKH II individuals [1,25,34,44,71,72,73,74]. As larger and more specialized studies using Whole-Exome Sequencing techniques emerge, other chromosomal locations and a clearer association between either MRKH type I or MRKH type II and a specific gene variant may be identified.

## 5. Conclusions

The genetic causes of MRKH syndrome remain elusive. Although some cases are familial, most cases are sporadic. In this study, we summarized and analyzed the most frequently reported genetic defects associated with MRKH syndrome in the available literature. The most reported chromosomal regions and the possible genes implicated are 1q21.1 (*RBM8A* gene), 1p31-1p35 (*WNT4* gene), 7p15.3 (*HOXA* gene), 16p11 (*TBX6* gene), 17q12 (*LHX1* and *HNF1B* genes), 22q11.21, and Xp22.

As there is a wider adoption of WGS techniques in MRKH studies, it is likely that, in the future, more genes and genetic regions will be identified. This information is particularly important because it can help clinicians associate clinical features in MRKH individuals with specific chromosomal regions and guide genetic counseling offered to patients and their families. Based on this knowledge, the prevention of the syndrome could also be possible through the development of appropriate gene therapy. However, larger cohort studies are necessary to elucidate the genetic basis of the syndrome.

## Figures and Tables

**Figure 1 children-09-00961-f001:**
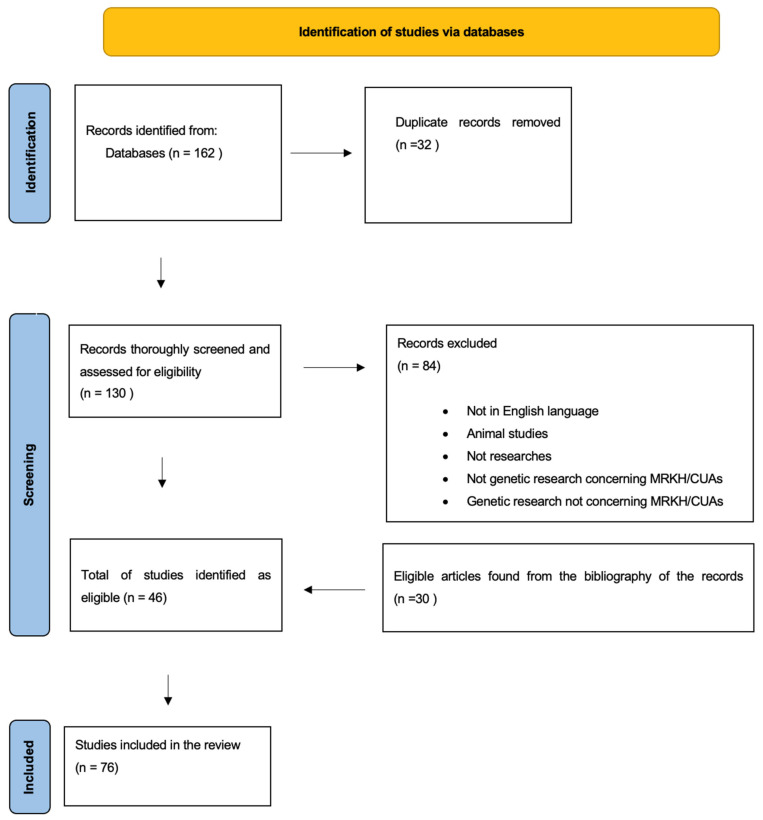
Identification process of the studies included in the Systematic Review.

**Table 1 children-09-00961-t001:** The most common chromosomal regions and genes associated with MRKH, their associated clinical presentation, animal studies of these genes, and phenotype of MRKH related to defects in these genes.

Chromosome Location	Suspected Genes Involved	Associated Syndromes	Non-Humans Study	Phenotype	References
1q21	*RBM8A*	TAR syndrome (thrombocytopenia, absence of the radius) [25,26,27,28]	Drosophila melanogaster: *RBM8A* encodes Y14 protein, which affects oocyte differentiation and determination of primordial germ cells [29]	Type I + II	[25,26,27,30]
16p11.2	*TBX6*	Autism spectrum disorders, neurological disorders, unaffected persons [28]	Mouse models: Deletion of *TBX6* presents skeletal (mainly vertebral) and urinary tract malformations [31,32]	Type I + II	[27,30,33,34,35,36,37,38]
17q12	*LHX1*	Anomalies in the embryogenesis, in body axis formation [28,39]	Mouse model: *LHX1* null mutant mice are characterized by absent uterus and oviducts [40] Mouse model: *LHX1* mutant mice had lack of kidneys and anencephaly [28,41] Mouse embryos with decreased *LHX1* activity had lower capacity of primordial germ cells (PGCs; [42])	Type I + II	[25,26,33,34,43,44,45,46,47,48,49,50,51,52]
	*HNF1B*	Renal cysts and diabetes [28]	Mouse models: Expression of *HNF1B* in MDs and in epithelial tissue of liver, pancreas, lungs and kidneys [53]		
22q11	Uncertain (*TBX1*)	DiGeorge or Velocardiofacial syndrome (heart defects, hypocalcemia, immunodeficiency, typical facial malformations, cognitive and behavioral disorders)		Type I + II	[25,26,27,33,54,55,56,57]

## Data Availability

Available upon request.

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
