# Peer review of "Identification of Genetic Causes in Mayer-Rokitansky-Küster-Hauser (MRKH) Syndrome: A Systematic Review of the Literature"

_children, 2022, doi:10.3390/children9070961_

Round 1
Reviewer 1 Report
Review of children-1781226
Identification of genetic causes in Mayer–Rokitansky–Küster– Hauser (MRKH) syndrome: A systematic review of the literature
Varvara Ermioni Triantafyllidi, Despoina Mavrogianni, Andreas Kalampalikis, Michael Litos, Stella Roidi, Lina Michala
General comments
This manuscript presents the systematic review of the genetic causes associated with MRKH syndrome and Congenital Uterine Anomalies. The main research findings of this paper will be important for the full understanding of these syndrome.
In my opinion, a substantial revision is needed to make this manuscript suitable for publication.
I have found a few issues that, once addressed, will improve the manuscript.
Specific comments
1. How did the authors define the suspected genes involved?
2. Are there any differences in the suspected genes between Type1 and Type2?.
3. Gene names should be replaced with italic.
4. The term “mutation” is not applicable in clinical.
Current guidelines of authorative organizations now also recommend to use neutral terms like “variant” and “change” only (see http://varnomen.hgvs.org/bg-material/basics/).
5. In discussion, what the differences between table 2 and lists of chromosomal regions?
6. In discussion line 243, what the meaning of “18q.1”? I did not follow this chromosomal region.
7. The authors lost two thirds of cases (84/130 records). I guess type 1 patients were not performed further studies and were not reported. The authors need to describe its limitations.
8. Eligible articles found from the bibliography of the records (n = 28) would be added in supplementary table.
9. The size of "," between references in the table is too large.
Reviewer 2 Report
Dear authors the manuscript is well-written, good references.
You highlighted the main genetic mutations for Rokitansky syndrome presented in litarature.
